# Exploring the Biomedical Frontiers of Plant-Derived Nanoparticles: Synthesis and Biological Reactions

**DOI:** 10.3390/pharmaceutics16070923

**Published:** 2024-07-11

**Authors:** Selvaraj Barathi, Srinivasan Ramalingam, Gopinath Krishnasamy, Jintae Lee

**Affiliations:** 1School of Chemical Engineering, Yeungnam University, Gyeongsan 38541, Republic of Korea; 2Department of Horticulture & Life Science, Yeungnam University, Gyeongsan 38541, Republic of Korea; 3Institute of Biomedicine, University of Turku, FI-20520 Turku, Finland

**Keywords:** biomedical, nanotechnology, plant extracts, cancer, toxicity, drug delivery system

## Abstract

As contemporary technology advances, scientists are striving to identify new approaches to managing several diseases. Compared to the more popular physiochemical synthesis, the plant-derived combination of metallic nanoparticles using plant secondary metabolites as a precursor has a number of benefits, including low expenses, low energy consumption, biocompatibility, and medicinal usefulness. This study intends to explore the impacts of using plant-derived synthetic materials including metallic nanoparticles (NPs), emphasizing the benefits of their broad use in next-generation treatments for cancer, diabetes, Alzheimer’s, and vector diseases. This comprehensive analysis investigates the potential of plant-derived remedies for diseases and looks at cutting-edge nanoformulation techniques aimed at addressing the function of the nanoparticles that accompany these organic substances. The purpose of the current review is to determine how plant extracts contribute to the synthesis of Silver nanoparticles (AgNPs), Gold nanoparticles (GtNPs), and platinum nanoparticles (PtNPs). It provides an overview of the many phytocompounds and their functions in biomedicine, including antibacterial, antioxidant, anticancer, and anti-inflammatory properties. Furthermore, this study placed a special focus on a range of applications, including drug delivery systems, diagnostics and therapy, the present benefits of nanoparticles (NPs), their biomedical uses in medical technology, and their toxicities.

## 1. Introduction

Because of their therapeutic qualities, extracts from plants have been utilized as traditional medicine for ages. These extracts come from a variety of plant components, including fruits, flowers, stems, roots, and leaves. The diverse array of bioactive compounds present in plant extracts has sparked interest in their biomedical application [1]. These compounds exhibit a wide range of pharmacological activities, making plant extracts valuable in the development of novel drugs and therapeutic interventions. Nevertheless, more than 70% of newly developed medications have low water solubility, which limits the drug’s ability to be absorbed following oral administration [2]. Some of the factors that contribute to the failure of clinical trials include the ingredient’s poor solubility, poor stability caused by gastric and colonic acidity, gut microbes, inadequate absorption across the intestinal barrier inadequate metabolism due to the influence of poor active elimination mechanism, and first-pass metabolic effects. These limitations hinder the accessibility of the active components in natural products.

The field of nanobiotechnology has revolutionized the synthesis of nanoparticles of metal oxide and provided significant advantages over traditional physical and chemical processes [3]. The vast range of medical and biological uses of a plant-derived methods of green synthesis has attracted a lot of interest [4]. The example of Zinc oxide nanoparticles, or ZnONPs, has attracted a lot of interest from a variety of industries, including medicine, optics, electronics, packaged foods, and biocompatibility. These features also help to keep the product’s cost down. Furthermore, new drug delivery methods and transport proteins for herbal drugs should ideally meet specific needs, like delivering the drug as needed during therapy at a rate dictated by the body’s needs and delivering the active component of the herbal drug to the site of action. Numerous strategies have been used to improve gastrointestinal permeability, sustainability, bioavailability, and medication solubility. When creating innovative drug carriers and delivery methods, nanocarriers have drawn a lot of interest. Encasing naturally occurring plant metabolites within a biocompatible and biodegradable nanoparticle is one way to combat this issue. Using a plant-derived green synthesis technology offers several advantages over traditional techniques [5]. To begin, it may be claimed that virtual procedures are simpler, have better safety safeguards, are more sustainable, and take a more environmentally conscientious approach than traditional chemical and physical alternatives. Furthermore, biologically synthesized nanoparticles produced by biological reactions have substantial biological features, making them suitable for a wide variety of biomedical uses in various fields [6]. Previous study has demonstrated that the antibacterial activity of plant-derived nanoparticles (NPs) produced using extracts from plants are superior to those of traditional medications in treating illnesses [7].

These nanoparticles have also been demonstrated to have anticancer qualities and show potential as agents that both scavenge free radicals and exhibit antioxidant activity [8]. Innovative treatments for diseases with malaria and infections of the urinary tract may be developed by plant-mediated green synthesis [9]. Moreover, the use of plant-mediated nanoparticles produced green has demonstrated significant promise for wound healing. The primary efficacious properties of these compounds stem from their antibacterial properties, particularly when included into formulations at the nano- and microscale [10]. These entities may also serve as biological sensors for the identification and evaluation of biomarkers linked to different clinical conditions. Green-produced ZnO nanoparticles have better biocompatibility and more biomedical properties than traditionally made ZnO nanoparticles, which makes them ideal for use as antibacterial substances and cancer-fighting medications. Additionally, it has been demonstrated that these nanoparticles may be used in sensing and drug delivery applications [11]. Finally, the development of nanobiotechnology has resulted in a paradigm change in the synthesis of metal oxide nanoparticles. When compared to traditional approaches, the use of plant-derived methodologies in green synthesis processes has proven to be particularly beneficial. Numerous studies have demonstrated that a number of nanoparticles have attracted interest from the scientific community due to their remarkable biocompatibility, low cytotoxicity, and affordability [12].

## 2. Combination of Plant Extracts and Nanoparticles

Plant extracts contain various bioactive compounds that have been utilized for centuries in traditional medicine. These extracts have gained significant attention in modern biomedical research due to their diverse pharmacological properties. The synthesis of plant extract–nanoparticle hybrids for biomedical applications combines the advantageous properties of both plant extracts and nanoparticles, leading to enhanced therapeutic potential [13], such as antioxidants, antimicrobials, and anti-inflammatory agents (Figure 1). When combined with nanoparticles, which can offer controlled release and targeted delivery of therapeutic agents, synergistic effects can be achieved, enhancing the overall efficacy of the treatment. Both plant extracts and nanoparticles are generally biocompatible and can be well tolerated by the human body [14]. This makes them suitable candidates for biomedical applications, including drug delivery, imaging, and tissue engineering. By shielding plant-derived bioactive chemicals from deterioration and promoting their uptake and distribution inside the body, nanoparticles can increase the bioavailability of these substances. This may result in the delivery of medicinal medicines to specific tissues or cells more effectively [15]. Therapeutic substances can be delivered precisely by targeting certain tissues, cells and organs with the use of functionalized nanoparticles. Plant extracts can be conjugated with nanoparticles to maximize therapeutic efficacy and minimize off-target effects while achieving focused medication delivery [16].

The toxicity of pharmaceuticals may be reduced by combinations of plant extracts and nanoparticles that permit the administration of lower doses while preserving therapeutic advantages. This may lessen the negative side effects brought on by greater medication concentrations [17]. Therapeutic compounds can be designed into nanoparticles such that they release in a regulated way either through sustained release kinetics or in reaction to certain stimuli (such as pH, temperature, or enzyme activity). Both patient compliance and the drug’s pharmacokinetics can be enhanced by this controlled release profile [18].

Plant extracts and nanoparticles can have multifunctional characteristics that allow them to perform several functions in a single composition [19]. For instance, they may be made to have antibacterial activity, transport medicinal drugs, and provide contrast for imaging at the same time, making them adaptable for a range of medical uses. In general, the creation of hybrids of plant extracts and manufactured nanoparticles shows potential for improving medicinal therapies by combining the complementing qualities of both materials. It is anticipated that further study in this field will result in the creation of novel treatment approaches with improved safety, effectiveness, and specific delivery capabilities

Green-synthesized nanoparticles provide a potentially beneficial avenue for the advancement of biomedical treatment techniques that are both effective and ecologically sustainable [20]. Numerous research groups have looked at the green production of nanoparticles of metal oxides as a possible substitute for traditional chemical and physical manufacturing methods. Notwithstanding their method of manufacturing, nanoparticles have raised concerns about potential environmental consequences. Their small size and unique traits might lead to different interactions with animals and surroundings [21]. Thus, to offer important insights into this cutting-edge research area, this review thoroughly examines the present state of plant-mediated synthesis, its recent achievements, the obstacles it faces, and its prospects.

## 3. Biomedical Applications of Plant-Derived Nanoparticles

Plant-derived nanoparticles hold significant promise for several treatments due to their unique properties, including biocompatibility, low toxicity, and the presence of bioactive compounds with many anti-disease properties. Here, we explore how these nanoparticles are being investigated for major disease treatments:

### 3.1. Cancer Therapy

As plant-derived nanotechnology targets a particular spot, it has drawn interest in the medicinal and diagnostic fields. Drug molecules, such as peptides, nucleic acids, and tiny medicines, either bind or encapsulate the plant-derived nanomaterials in therapies. This creates a therapeutic entity the size of a plant-derived nanoparticle that targets malignant cells without adhering to healthy cells. The medicinal substances from the nanoparticle travel to the location and carry out the therapeutic activity after they have targeted the malignant cell. In the same way, plant-derived nanoparticles are used in diagnostics to recognize tumor cells. Here, variously shaped nanoparticles, such as nanoshells and nanotubes, are created. After adhering to a nanoshell, the antibody detects the tumor cell and sends out a signal. Because of their special physicochemical characteristics, metallic nanoparticles are the perfect materials for the therapeutic targeting of illnesses like cancer. Many techniques, such as wet chemical synthesis, pyrolysis, the method known as hydrothermal precipitation, co-precipitation, the sol–gel process, microemulsion, sonolysis, and reduction, can be used to synthesize them.

When creating targeted therapies, it is preferable to use both chemical and physical characteristics, such as the enhancement of enzyme activity and fluorescence, luminescence, and surface plasmon resonance, among others. These plant-derived nanoparticles also show a high surface area to volume ratio, which increases the possibility that they will be a good fit for coating with different medications and tiny molecules. These plant-derived nanoparticles are favored for targeted cancer therapies due to their surface functionalization, which also results in fewer side effects. Additionally, nanoparticles made of metal have been connected to cancer imaging and diagnosis [11]. Moreover, additional phenomena like photothermal and the hyperthermia effect might be linked to the killing of cancer cells caused by plant-derived nanoparticles (Table 1). The shape, morphology, and topographical characteristics of the nanoparticles determine their capacity to trigger these activities. Metal nanoparticles (NPs) exhibit biocompatibility, possess innate anticancer properties, do not build up within the body, and may be tailored to form combinations with other NPs. They can also act as different radioisotopes and fluorescent dyes for imaging reasons [22].

Numerous biological uses of plant-derived nanoparticles exist, such as gene transport, antifungal applications, anticancer applications, medication delivery, thermal ablation, and augmentation of radiotherapy [58]. To target different cell types, they are functionalized with a variety of functional groups, such as antibodies, peptides, DNA, and RNA, and biocompatible polymers. One example of a nanostructure used in breast cancer treatment is made of branching gold shells. Furthermore, cancer cells were treated using plant-derived nanoparticles. Metallic nanoparticles are among the most often used agents in cancer theragnostic. Compared to conventional cancer treatments, they provide several advantages, such as fewer adverse reactions and a lower rate of drug resistance [59]. These NPs have recently been modified with aptamers, silica, DNA, photosensitive substances, fluorescent molecules, and photoluminescence to improve their suitability for imaging, diagnostics, and other applications.

The application of plant-derived nanoparticles in cancer theragnostics has gained significant attention due to their unique physicochemical properties. Plant-derived nanoparticles, such as gold, silver, and platinum, have demonstrated potential in both cancer diagnosis and therapy. In diagnosis, these nanoparticles can be functionalized with targeting molecules and imaging agents to specifically bind to cancer cells, allowing for precise detection through various imaging modalities such as MRI, CT scans, and optical imaging. This targeted approach enables early stage cancer detection and precise visualization of tumors. In therapy, metallic nanoparticles can be utilized as delivery platforms for therapeutic agents, including chemotherapy drugs and nucleic acid-based treatments. Their large surface area allows for efficient drug loading, and their unique optical and electromagnetic properties can be harnessed for photothermal and photodynamic therapy to selectively destroy cancer cells.

### 3.2. Alzheimer’s Disease

The blood–brain barrier (BBB) is a finely tuned network of endothelial cells and blood arteries that prevents undesirable substances from entering the brain. This barrier searches for treatments for neurodegenerative illnesses incredibly difficult and complex. The advancement of nanotechnology presents encouraging opportunities to address this problem. It is expected that developments in nanotechnologies in neurological research will have a major influence on the advancement of innovative therapeutically techniques. Since they may stimulate, react to, and affect target cells in addition to tissues to protect the intended physiological reactions while minimizing undesired results, protein-sized nanoparticles are produced with a broad variety of medicinal uses. The degeneration of neuronal subtypes over time characterizes a class of chronic, perhaps genetic diseases known as neurodegenerative disorders (NDs). 

Alzheimer’s disease (AD) exerts the greatest impact on society in the twenty-first century [60].

The ensuing immunological stimulation of the central nervous system (CNS) causes neurodegenerative disorders, which disproportionately affect the public and health sectors. When paired with additional mechanisms like eliminating necrotic cells and avoiding neurotropic viral infections, immune activation can aid in regeneration and repair; yet, it can also result in immune-mediated illnesses, ischemia, infections, and neurodegeneration. The gradual loss of neurons and axons in the central nervous system (CNS) results in abnormalities in cellular function and, eventually, cellular death [61,62]. This process is referred to as “neurodegeneration”.

*Allium sativum*, a member of the family *Amaryllidaceae* and a plant that has been used for medicinal purposes for thousands of years, is another plant-based remedy. It has been used to treat atherosclerosis, hyperlipidemia, coagulation, high blood pressure, Alzheimer’s disease, cancer, and diabetes. S-allyl cysteine (SAC), a component of aged garlic extract (AGE) that has been extensively studied, is one of its main ingredients. There is evidence of indirect as well as direct antioxidant action by SAC. Along with minimizing DNA fragmentation and lipid peroxidation, it also lowers nitration and oxidation. SAC preserves dopamine levels, prevents oxidative damage, and causes lipid peroxidation in Parkinson’s disease (PD) models using 1-methyl-4-phenyl pyridinium (1-OHDA) and 6-hydroxydopamine (6-OHDA) [63]. The golden-colored turmeric is produced by the *Zingiberaceae* plant *Curcuma longa* and has been historically used medicinally for millennia. Curcumin can bind to Aβ peptides, which stops the formation of new deposits of amyloid and speeds up the de-aggregation of existing amyloid deposits, according to a number of ongoing studies in Alzheimer’s disease patients [64]. Another well-known and widely used phytomedicine that has been historically employed as a memory-enhancing medication is *Salvia officinalis*, which is part of the *Lamiaceae* family. According to research conducted to date, it could be able to prevent or lessen dementia symptoms. Cognitive function significantly improved in 11 individuals with moderate signs of AD after receiving essential oil derived from *S. aureus* orally [65]. Thus, the use of plant-derived nanomaterials opens up new avenues for treating AD by improving the functional properties of the blood–brain barrier. In order to create scaffolds that provide biological, mechanical, and structural support for penetration through the blood–brain barrier, nanomaterials seem like a viable starting point. Overall, it has been shown that scaffold properties such porosity, biocompatibility, degradability, and 3D design are essential components of biomaterial engineering.

### 3.3. Diabetes

Diabetes mellitus (DM) is the most difficult undiagnosed illness of the twenty-first century, encompassing a chronic set of metabolic illnesses. A more exact and accurate substitute for the diagnosis and treatment of diabetes mellitus may be offered by the application of nanotechnology in the shape of nanoherbal medications [66]. Reverse pharmacology has been used to cure a number of ailments by using several plant compounds in traditional medications. One of the most promising treatments for diabetes control is the use of scenario-based medications supported by current scientific evidence [67]. Consequently, many diabetic patients have been interested in this extra or alternative therapy using herbal drugs [68,69,70]. Therefore, there is an urgent need for a new, innovative medication using cutting-edge modern technology, and scientists are looking for a new medication using green or plant-associated nanoparticles. Natural source-derived silver nanoparticles have been demonstrated in several studies to possess a variety of beneficial effects of antidiabetic qualities [71]. Notably, both the avoidance and management of diabetes and its aftereffects depend heavily on the discovery of a medication with several bioactivities, such as antioxidant, antihyperglycemic, and antihyperlipidemic properties.

More and more research points to the effectiveness of natural compounds in treating diabetes in people with the condition. Almost half of all pharmaceuticals on the market today are derived from sources that are natural or are extracted using natural chemicals [72]. The optimum treatment pattern can lead to the creation of theragnostic nanomedicine, an expanding discipline. It offers several benefits, including the ability to load phytoconstituents as medications onto nanocarriers, which is advantageous for both therapeutic and imaging purposes. Theragnostics is a revolutionary strategy that uses a single system to regulate the combination of tailored therapies and diagnostic testing. Theragnostic drugs are coupled with nanomedicine to transport, image, and carry out therapeutic actions all at once [73]. Treatment for resistance to insulin in diabetes type 2 mellitus using a plant-derived, natural, biocompatible insulin sensitizer that works with a nanodelivery technology. To produce a self-assembling micellar form for oral administration of polygalacturonic acid-laden oleanolic acid (PGAOA), the herbal insulin sensitizer oleanolic acid (OA) was loaded onto plant-derived polygalacturonic acid. According to earlier in vitro as well as in vivo experimental investigations, OA-loaded PGAOA micelles exhibit exceptional stability in overcoming gastrointestinal barriers, resulting in improved medication intestinal absorption, and remarkably prolonged plasma drug concentration maintenance [74]. Therefore, the T2DM rat model’s insulin resistance was effectively reversed by the nanotechnology of OA-loaded PGAOA micelles, and it also demonstrated a long-lasting impact on glucose regulation even after the medication was removed.

In Indonesia, common plants with antioxidant qualities are *Rhodomyrtus tomentosa* (haramonting) leaves and *Zanthoxylum acanthopodium* fruits [47]. Patients with diabetes mellitus have elevated glucose levels, which impact angiogenesis and ultimately the rate at which wounds heal. Fibroblast growth factor (FGF) in the epidermal tissue of the rats was utilized to measure the histological alterations of diabetic wound healing using the nanoherbal medications haramonting and andaliman. The results of the experimental investigation demonstrated that the thick basal membrane had been neatly structured, the epidermis had been covered by the epithelialization process, and the dermis was filled with a dense connective tissue that increased the amount of fibroblast cells. The use of the herbal medicines haramonting and andaliman encouraged the development and division of cells to repair the injured skin layer [75]. Additionally, the antidiabetic efficacy of the SNP was tested against STZ-induced diabetic experimental mice using a leaf extract from Pouteria sapota. This work has unequivocally demonstrated that administration of silver nanoparticles generated from *P. sapota* leaf extract results in a considerable drop in blood sugar levels. Furthermore, our investigation has validated the antidiabetic properties of this nanoparticle in both in vitro and in vivo investigations.

### 3.4. Mosquito-Borne Diseases

The bite of a female mosquito carrying an illness can transmit diseases carried by mosquitoes. Malaria, Influenza, the Zika virus, Dengue fever, filariasis of the lymphatic system, and tick-borne encephalitis are the principal illnesses spread by mosquitoes [76]. Mosquitoes of the genera Aedes, Culex, and Anopheles, which are extensively found in Asia, Africa, South America, and Europe, serve as significant vectors of the pathogens that cause these illnesses [77]. Over one million people die annually as a result of mosquito-borne diseases, which affect over 700 million people worldwide [78]. The most serious of them is malaria, which is spread to people by female Anopheles mosquitoes carrying the infection. Asia, South America, and the tropical regions of Africa are all affected by malaria [79]. 

As mentioned above, pharmacological, economic, and environmental sustainability challenges have made the use of synthetic insecticides (such as carbamates, organochlorines, organophosphates, and pyrethroids) unsatisfactory [80]. Their high prices, the worldwide rise in vector species’ resistance, the biotic extension of hazardous byproducts in the food chain, their detrimental impacts on humans as well as non-target creatures, and their non-biodegradable nature are some of these worries and challenges [81]. In light of this, the creation of safer, more economical, environmentally friendly, biodegradable, more effective, and target-specific plant-derived pesticides has emerged as one of the most successful substitute strategies for controlling mosquito populations. Approximately two thousand plant species belonging to different plant families have been demonstrated to possess ovicidal, insecticidal, pupicidal, adulticidal, and possibly repellent qualities against different types of mosquitoes, according to the literature [76]. These plants have noteworthy insecticidal effects because they vary depending on several factors, such as species, origin and seasonal variation, ages (senescent, mature, or young), parts utilized (leaves, bark, stems, roots, etc.), extraction technique and solvent polarity, photosensitivity for some of their compounds, and the mosquito vector species and stages of development.

In contrast, it was shown that fewer than 40% of research about the mosquitocidal assessment of botanical-derived pesticides used water as a solvent for extraction. Even though these aqueous extracts are thought to be environmentally benign and hardly hazardous to humans (and other animals), only half of them have a fifty percentage-point lethal concentration (LC50) less than 30 ppm. Moreover, it lacks the resources to be really costly to make and use for those who are impoverished anywhere in the globe. Citations include (i) ethanol–water extracts of *Centella asiatica*, *Cassia fistuala*, and Artemisia annua [82], (ii) aqueous extracts of *Carica papaya*, *Murraya paniculata*, and *Cleistanthus collinus* [83], and (iii) steam-distilled extracts of *Allium sativum*, *Artemisia vulgaris*, *Argemone mexicana*, *Platycladus orientalis*, *Tradescintia zebrine*, *Paullinia claviger*, and *Lantana camara*.

Notably, several plant-derived essential oils also demonstrated strong insecticidal and repellant properties for mosquitoes. One significant drawback, nevertheless, is that distributing them in an aqueous environment necessitates the employment of synthetic surfactants [84]. The primary benefit of the plants listed above, together with the corresponding homemade medicines, is the physical restriction of mosquito penetration into the home, which is beneficial in the fight against malaria and other diseases spread by mosquitoes. [85]. However, a number of problems have prevented a few promising plants with outstanding toxicological features toward mosquito vectors from producing commercially viable products. These problems include the small determination of their insecticide impacts, the drawn-out and expensive process of producing plant-derived insecticide products, and the insufficient strength of some of their bio-active plant-based substances in the environment [85].

In light of the aforementioned, it is imperative to look for suitable formulations and/or stabilizing techniques to increase the efficacy and durability of plant-derived pesticides and goods made from them [86]. The most promising methods to achieve this goal are to use plant extracts and particular elements such as reducing and stabilizing representatives during the combination of metal oxide and green metal nanoparticles. Fascinatingly, this technique is called “green synthesis of metallic nanoparticles”, and it involves converting bioactive materials, including plants, plant extracts, microorganisms, and enzymes, into different metallic nanoparticle forms [87]. Green synthesis is an ecological technique that does not involve handling hazardous chemicals; therefore, it protects the environment.

It is necessary to use a “green” solvent, an eco-friendly reducing agent, and a non-toxic stabilizing chemical in that order [88]. Moreover, green nanosynthesis allows for the reliable use of minimal-cost agricultural wastes (such as fruit peels and removal method remains), invertebrate byproducts (such as crab shell chitosan), and fungal extracellular filtrates as a consistent source of reducing agents [89]. To create green synthesized plant-derived metallic nanoparticles, diverse concentrations of fresh plant extracts (or other naturally occurring sources of reducing agents) are mixed with diverse concentrations of metal precursor solutions under different volume proportions and conditions for reaction (pH, temperature, stirring speed) [90].

### 3.5. Drug Delivery System

Plant-derived nanoparticles have garnered attention for their potential application in targeted drug delivery due to their unique properties. These nanoparticles, derived from plant sources such as polyphenols, flavonoids, and other phytochemicals, offer an exciting opportunity to revolutionize the field of medicine. Researchers have been actively investigating the usage of plant-derived nanoparticles for delivering an extensive variety of therapeutic compounds, including anticancer drugs, anti-inflammatory agents, and antimicrobial agents [91]. The biocompatibility and biodegradability of plant-derived nanoparticles make them an attractive option for delivering drugs to specific target sites within the body (Table 2). Furthermore, their low toxicity profile offers the potential to minimize adverse effects, making them a safer alternative to conventional drug delivery systems [92]. In recent years, significant advancements have been made in the synthesis and functionalization of plant-derived nanoparticles, allowing for precise control over their size, shape, and surface properties. These developments have opened up new possibilities for optimizing drug loading and release kinetics, further enhancing the efficacy of drug delivery systems [93].

Moreover, the ability of plant-derived nanoparticles to be modified with targeting ligands and imaging agents holds great promise for enabling site-specific delivery and real-time monitoring of drug release [93]. This capability is instrumental in improving the therapeutic index of drugs and facilitating personalized medicine approaches. As we delve deeper into the potential of plant-derived nanoparticles for drug delivery, it becomes evident that their multifaceted advantages have the potential to reshape the landscape of pharmaceutical interventions [100]. In the following sections, we will delve into the specific advancements and applications of plant-derived nanoparticles across different therapeutic areas, highlighting their transformative impact on the field of medicine (Figure 2).

The primary ingredient of *Curcuma longa*, curcumin, has a high metabolism and is lipophilic, which makes it a poor bioavailability candidate for an anticancer treatment [101]. Numerous investigations have shown that curcumin nanoliposomes made by the ethanol injection technique, which combine sodium hyaluronate with trimethyl CS to produce polymer-glycerosomes, may efficiently transport curcumin to the lung, hence increasing its therapeutic index [102]. In 2018, Zhang et al. created liposomal curcumin powdered inhalers (LCDs) and curcumin liposomes for pulmonary delivery-based inhalation therapy of primary lung cancer. Because curcumin liposomes were administered via the pulmonary route in this trial, metabolic restrictions could not apply. Moreover, liposomal formulation greatly contributes to an improved anticancer impact by enhancing the water solubility of curcumin [103].

The production of nanomaterials and their use in diagnosis and treatment in medicine have advanced significantly in recent years because of advances in nanotechnology. However, nanomaterials are still not applied to plants in significant amounts. Based on current research and applications, more investigations are required to improve the production and bio-functionalization of nanomaterials. It is plausible to argue that several barriers that fall into the following categories are preventing the full potential and advantages of using nanomaterials in agricultural sciences and agriculture: (i) the necessity of creating and synthesizing safe natural flavors; (ii) the significance of bioactive ingredients; (iii) the ignorance of extraction procedures; and (iv) the absence of the interdisciplinary methods necessary for developing and putting into practice uses of nanotechnology in plants.

## 4. Potential Anticancer Mechanism of Action of Plant-Derived Nanoparticles

The anticancer properties of biosynthesized nanoparticles have been the subject of much investigation, although the precise processes behind this process are still unclear. Reactive oxygen species (ROS) are usually produced by plant-derived nanoparticles, and this results in cellular death. ROS causes cellular death by changing signal transduction pathways [104]. Phytoconstituents can cause molecular and cellular changes that are either stimulating (mostly of mitochondrial pathway and cell death) or inhibiting (primarily of anticancer properties such as cell viability, tumor assault, growth, proliferation, and development). NPs can obstruct the actions of endocytosis, phagocytosis, and pinocytosis in human cells [105]. NPs can transport ions that can enter the nucleus, cause DNA damage or hypermethylation, or halt the cell cycle in cancerous cells (Figure 2). Furthermore, the reasons for NPs’ detrimental effects on cellular viability include the downregulation of antiapoptotic genes like Bcl2, the generation of reactive oxygen species (ROS), mitochondrial autophagy among fission, and processes that eventually lead to cell death by apoptosis [106]. For instance, it has been demonstrated that AgNPs can be cytotoxic to mammalian cells via a number of mechanisms [107] such as the breakdown of cells dependent on energy cellular roles and reduced DNA replication due to free silver ion approval, (b) the generation of free radicals and reactive oxygen species (ROS), and (c) damage to cell membranes resulting from direct interaction with AgNPs [108].

Through internal and external pathways, nanomaterials can trigger apoptotic signaling. When the intrinsic route triggers apoptosis, the production of ROS depolarizes the mitochondrial membrane, releasing cytochrome *c* into the cytosol. Pro-apoptotic proteases in the extrinsic route of apoptosis are subsequently activated by cytochrome *c*, initiating the caspase-9/3 apoptotic cascade [109]. Zheng et al., reported that AuNP treatment generated reactive oxygen species (ROS) in A549 cells, which may trigger oxidative stress-induced apoptosis [110]. Additionally, Bcl2 and Bid were downregulated, while Bax, caspase 3, and Beclin 1 were overexpressed, with IC50 values ranging from 20 to 25 µg/mL. In an additional in vitro study on lung cancer, AgNPs revealed a strong inhibitory impact on H1299 cells, with an IC50 of 5.33 ± 0.37 µg/mL [111].

Conversely, the potential of certain green-produced nanoparticles to fight cancer may be related to the inherent properties of the proteins or phytochemical components in the plant extract. TiO_2_NPs were made by Sharma et al. from Rheum emodi roots, which are known to have medicinal properties since they contain aloe-emodin, emodin, and chrysophanol anthraquinones [112]. Using an excellent diffusion method, the cytotoxic effect of these NPs was examined. The results demonstrated the highly active role of biosynthesized TiO2NPs as a nanomedicine that may be employed to treat tumor cells while protecting healthy cells. 

## 5. Green Manufactured Nanoparticles’ Toxicity

Green-synthesized nanoparticles have a lot of potential in the biomedical field; however, there are firm unfavorable health consequences connected to their usage. Given that one of the main issues with the constraints of employing nanoparticles in medical uses is toxicity. As a result, research has to be conducted on the consequences of increasing nanoparticle exposure on humans, animals, and the surrounding environment, as well as any possible risks for acute and long-term toxicity [113]. Green-generated nanoparticles (NPs) have varying degrees of toxicity to biological systems depending on their size, shape, surface chemistry, elasticity, composition, and the targeted ligand. Furthermore, reactive oxygen species were created when metal-containing nanoparticles were introduced to human lung epithelial cells. These species can result in oxidative stress and damage to cells [114]. This article discusses the possible risks and hazards that green-manufactured nanoparticles may pose to cells, both in vivo and in vitro.

According to a recent study on the negative impacts of plant-mediated AgNPs, these substances exhibited a strong anti-lung cancer effect, possibly with minimal danger to normal cells (Figure 3). For example, Kanipandian et al. use in vivo research to investigate the harmful effects of AgNPs produced by *G. hirsutum* on the biological system [43]. Interestingly, the mice treated with AgNPs showed no pathogenic evidence, suggesting that AgNPs do not endanger vital organs. Kummara et al. investigated the cytotoxicity of Aza-dirachta indicia-mediated AgNPs on the NCI-H460 non-small cell lung cancer cell line [115]. It was discovered that 50% of the malignant cells were killed at a concentration of 120 × 10^−6^. When it comes to chemical AgNPs, cytotoxicity has been demonstrated in typical cells at 120 × 10^−6^ concentrations. Normal cell lines are more biocompatible with biosynthesized AgNPs since they are not subjected to an acidic pH. These results were confirmed by the observation that AgNPs made from *Olax scandens* leaf extract had significantly more killing effect on normal cell lines than in cancer cells [116].

Another study examined the in vitro cytotoxicity and cell survival of the produced AgNPs using *H. cannabinoid* seed extract against a human lung cancer cell line (A549) and mouse embryonic fibroblast cells (NIH3T3). According to the findings, AgNPs had a dose-dependent moderate toxicity effect on normal cells (NIH3T3), with an optimal dosage of 1000 μg/mL showing a 32.77% inhibition of cell growth [94]. After being exposed to synthetic AgNPs comprising up to 14.08 mg/L for 24–48 h, *P. reticulata* showed no signs of toxicity from AgNPs generated from *P. daemia* and *P. rubrasis* latex [117]. Similar to this, green-produced AuNPs can cause oxidative stress and inflammation in cancer cell types, producing ROS that promote DNA damage and ultimately cause cell death [118].

Anand et al. investigated the cytotoxicity of AuNPs derived from *M. oleifera* in converted A549 and regular healthy peripheral cells [34] using MTT analyses.The results showed that A549 cells are considerably more vulnerable to AuNPs cytotoxicity than normal. In a different investigation, the toxicology of ZnONPs biosynthesised from *Cycas pschannae* leaf extract was assessed using zebra fish embryos. The results indicated that *Daniorerio* had a low embryo mortality, suggesting a greater level of biocompatibility at 20 μg/mL. Dose-dependent toxicity was seen when IONPs were administered to human lung cell lines (L132) that were healthy and lung cancer cell lines (A549). On the other hand, cancer cell lines (A549) were more toxic to IONPs made from *Phyllanthus emblica* fruit extract than normal cell lines (L132) [119]. PtNPs made with *M. royleanus* leaf extract have been shown to have little cytotoxicity towards normal cells and anticancer effectiveness against A549 tumor cells [120]. The plant-mediated nanoparticles’ phytobiochemical coatings are readily released in acidic tumor settings, which enhances the nanoparticles’ anticancer properties. Because biocompatible phytoconstituents are present, biological production of NPs is favored to minimize toxicity [121]. As a result, it is far safer to create NPs made from plants than from chemicals [122]. These investigations all attest to the significant anti-lung cancer action of plant-mediated NPs as well as their low toxicity to healthy cells. In the near future, such results may offer a fresh approach to using NPs in lung cancer clinical trials. Due to their high selectivity and low toxicity, green-synthesized nanoparticles that are also ecologically benign may be strong future prospects for biological uses, even in light of their costs.

## 6. Recent Achievements of Plant Synthesized Nanoparticles

The study of biomedicine has benefited greatly from the recent successes of plant-derived nanoparticles, which demonstrate their promise for a wide range of applications. Their contribution to the creation of cutting-edge methods for healthcare and illness treatment is one noteworthy development. Growing evidence has shown that plant-derived nanoparticles can be useful instruments for targeted medication delivery [123]), which might revolutionize the way that many illnesses are treated. More accurate and successful therapeutic interventions have resulted from their special capacity to encapsulate medicinal chemicals and release them at certain cellular targets. By reducing adverse effects and increasing medicine efficacy, this customized delivery method eventually improves patient outcomes [59].

Plant-derived nanoparticles have also demonstrated potential in the fields of regenerative medicine and tissue engineering. As of their bioactive qualities and biocompatibility, they are excellent choices for the development of novel biomedical treatments meant to restore and regenerate tissues that have been damaged. With the potential to transform the creation of sophisticated biomaterials and scaffolds for tissue regeneration, these nanoparticles might provide patients suffering from degenerative diseases or traumas with fresh hope.

## 7. Future Direction

One of the most exciting developments in the field of biomedical treatment is the application of nanotechnology for targeted drug delivery, which will influence pulmonary treatments going forward. Although the highly effective bioactive chemicals originating from plants are well recognized, their limited therapeutic value stems from their low stability and solubility. The problem of making these phytocompounds more soluble and shielding them from the digestive tract’s enzymatic degradation has been solved by using a variety of nanoformulation techniques. Nanoformulations, including nanoparticles, liposomes, and nanofibers, provide a number of benefits for pulmonary delivery of bioactive compounds obtained from plants. Moreover, nanocarriers may be tailored to strengthen their attraction for specific target cells throughout the pulmonary system, which can improve the effectiveness of drugs while reducing systemic adverse effects.

Improved bioavailability is therefore essential to increasing the curative efficacy of plant-derived medications, and several studies have demonstrated the synergistic benefits of combining several plant-derived substances. The benefits of synergism can be realized by co-encapsulating several bioactive substances on a platform made possible by nanoformulations. With the use of this approach, effective application customized with many treatments that address a variety of disease-related factors, including inflammation, constriction of the airways, and oxidative stress, may be possible. Advances in tailored medicine and nanotechnology are making it possible to tailor therapy to individual patient features. Researchers may design customized nanoformulations for each patient by utilizing patient-specific data, including genetics, biomarkers, and severity of illness. This approach may result in more targeted, potent drugs with less adverse effects and better therapeutic outcomes. Combination drugs may make it possible to create individualized treatment programs that are specific to the needs of each patient, enhancing the efficacy of care overall. Additionally, reducing the dosages of each particular medication in the combination regimen may lead to fewer side effects, enhancing patient comfort and adherence. To allow for a more targeted approach, future research might search for biomarkers that indicate patient reactions to certain combination medications. In order to assess the safety and effectiveness of combination therapy for several diseases, comprehensive clinical studies involving industry, academia, and regulatory bodies are necessary.

## 8. Conclusions

Utilizing as many bioactive substances derived from plants as is practical is one possible strategy to manage many illnesses more successfully. A completely new era of cancer, diabetics, and Alzheimer’s treatment has been ushered in by the advancement of plant-derived nano techniques. By using the potential of nanotechnology, we can go beyond the obstacles that have prevented the proper use of bioactive compounds originating from plants. In our review paper, we talked about the pharmacological and therapeutic uses of metallic nanoparticles that have been biosynthesized from diverse medicinal plants, as well as their applications in the fields of biomedical, drug delivery, nanomedicine, and diagnostics. The absence of harmful impurities makes medicinal plant nanoparticles ideal for use in therapeutics and medical sciences. Comparing the biosynthesized medicinal plant nanoparticles to their chemically and physically synthesized counterparts, the former have attained an appropriate degree of competency or biofunctionality.

## Figures and Tables

**Figure 1 pharmaceutics-16-00923-f001:**
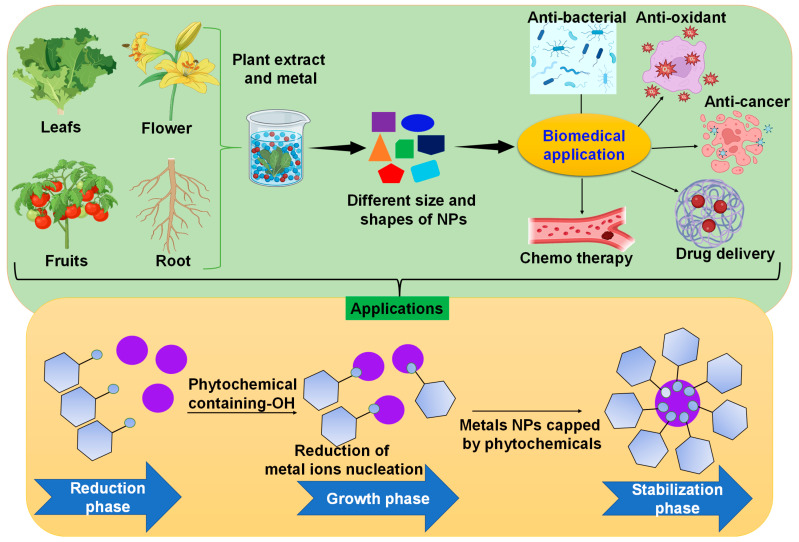
A proposed synthetic pathway for plant-mediated nanoparticle production and their advantages of green synthesis over traditional techniques.

**Figure 2 pharmaceutics-16-00923-f002:**
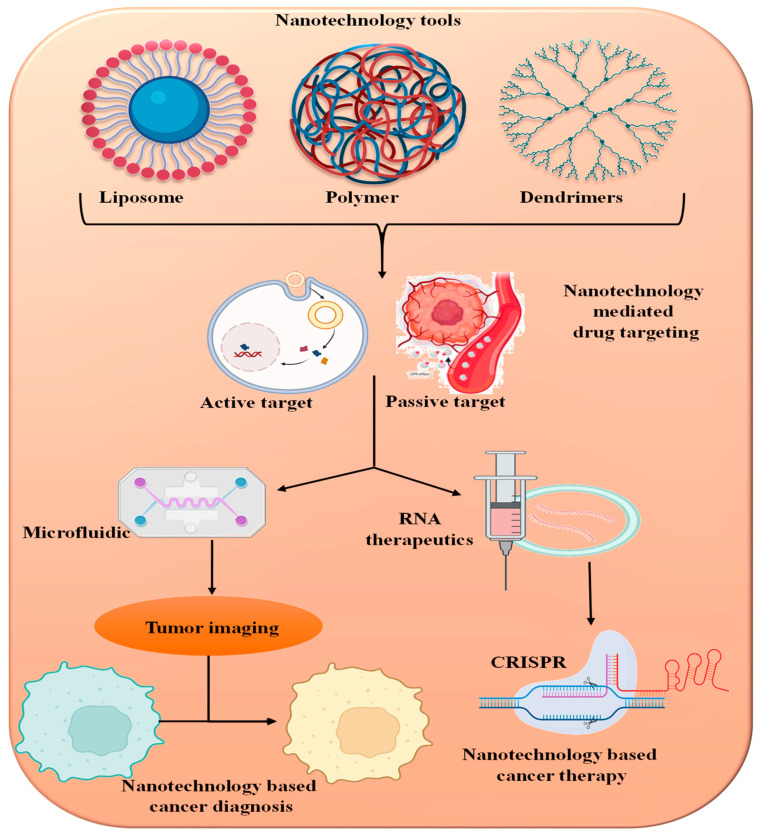
Nanotechnology tools for cancer detection and therapy. Different nanotechnological methods are used for targeted delivery of drug nanoparticles system.

**Figure 3 pharmaceutics-16-00923-f003:**
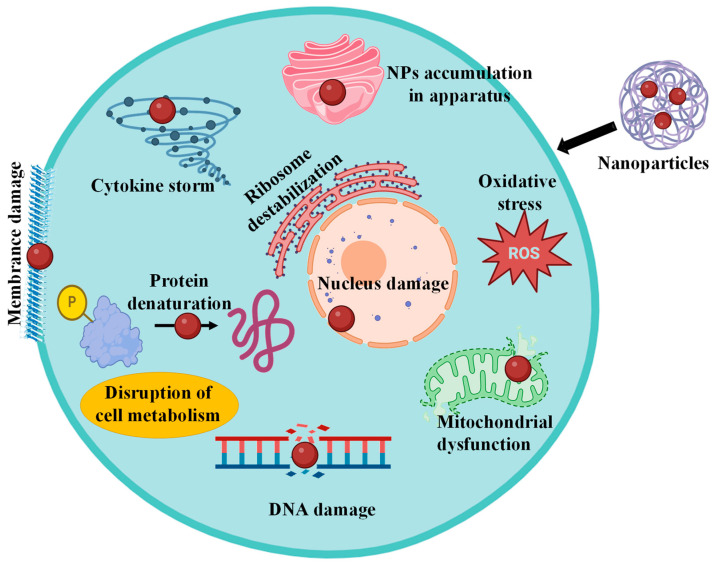
Diverse impacts of nanoparticles on cell.

**Table 1 pharmaceutics-16-00923-t001:** Types of planted-derived nanoparticles that showed anticancer potential.

Plant	Part Used	NP Size and Shape	Absorbance	Reference
**Platinum (PtNPs)**				
*Rosemary* and ginseng	Leaves	Spherical, 197.2 nm	122 nm	[23]
*Cacumen platycladi*	Leaf extract	Spherical 212.3 nm	214.4 nm	[24]
*Punica granatum*	Peel extract	Spherical, 16–23 nm	260 nm	[25]
*Barleria prionitis*	Leaf extract	Spherical, 2–20 nm	570 nm	[26]
*Antigonon leptopus*	Leaf stem and root extract	Spherical, 2–4 nm	200 to 800 nm	[27]
**Gold nanoparticles (AuNPs)**				
*Bauhinia purpurea*	Leaves	Spherical	560 nm	[28]
*Dodonaea viscosa*	Leaves	Spherical, 30–90 nm	480 to 670 nm	[29]
*Alternanthera bettzickiana*	Leaves	Spherical, 80–100 nm	520 nm	[30]
*Nigella arvensis*	Leaves	Spherical 3–37 nm	546 nm	[31]
*Piper betle*	Leaves	Spherical, 10 nm	540 nm	[32]
*Rabdosia rubescens*	Leaves	Ring, 130 nm	550 nm	[33]
*Moringa oleifera*	Leaves	Spherical, 203 nm	540 nm	[34]
*Indigofera tinctoria*	Leaves	Spherical, 19.73 nm	545 nm	[35]
*Euphrasia officinalis*	Leaves	Spherical, 49.72 ± 1.2 nm	558 nm	[36]
**Silver nanoparticles (AgNPs)**				
*Acorous calamus*	Rhizome	Spherical, 31.83 nm	421 nm	[37]
*Camellia sinensis* L.	Leaves	Spherical, 8.29 nm	451 nm	[38]
*Croton bonplandianum*	Leaves	Spherical, 32 nm	425 nm	[39]
*Allium sativum* L.	Bulb	Spherical 8.18 nm	449 nm	[36]
*Borago officinalis*	Leaves	Spherical 30 to 80 nm	442 nm	[40]
*Albizia adianthifolia*	Leaves	4–35 nm	448 nm	[41]
*Beta vulgaris*	Taproot	Spherical, Circular 5–20 nm	450 nm	[42]
*Gossypium hirsutum*	Leaves	13–40 nm/spherical	410 nm	[43]
*Chrysanthemum morifolium*	Longan peel	27.2 nm	430 nm	[44]
*Avicennia marina*	Leaves	10–20 nm/spherical	420 nm	[45]
*Derris trifoliate*	Seed	16.05 ± 5.0 nm	419 nm	[46]
*Curcuma longa* L.	Rhizome	Spherical, 6.06 nm	447 nm	[38]
*Cymodocea serrulata*	Leaves	Spherical, 29.28 nm	420 nm	[47]
*Cymbopogon citratus*	Leaves	Spherical, 17–25.8 nm	435 nm	[48]
*Dendropanax morbifera*	Leaves	Polygon, 100–150 nm	548 nm	[49]
*Dodonaea viscosa*	Leaves	Spherical, 70–100 nm/	441–654 nm	[50]
*Zanthoxylum rhetsa*	Seed	Spherical, 10–68 nm	426 nm	[51]
*Syzygium aromaticum*	Fruit	Spherical, 5–20 nm	470 nm	[52]
*Panax ginseng*	Leaves	Spherical, 5–15 nm	420 nm	[53]
*Matricaria chamomilla*	Leaves	Spherical, 45.12 nm	430 nm	[54]
*Pinus roxburghii*	Fruit needles	Spherical, 80 nm	459 nm	[55]
*Ocimum americanum*	Leaves	Spherical, 48.25 nm	435 nm	[56]
*Punica granatum*	Peel	Spherical, 6–45 nm	450 nm	[57]

**Table 2 pharmaceutics-16-00923-t002:** Biomedical plant-based therapeutics and Precision Drug delivery system.

S. No	Plant-Derived Nanoparticles	Types of Drug Delivery	Target	Application	Ref.
1	Liposome	Capsaicin	Antimicrobial activity	A rise in antimicrobial activity	[94]
2	Liposome	Curcumin	Anticancer activity	Considerable cytotoxicity against MCF-7 cells and extended curcumin release enhanced antitumor effect	[95]
3	Grapefruit	Dox, Curcumin and Paclitaxel	Colon cancer	Improved antitumor effect	[96]
4	Phytosome	Naringenin	Acute lung injury	Increased pulmonary absorption of naringenin	[97]
5	Phytosome	Ginsenosides	Antioxidant activity	enhanced ginsenoside absorption and effectiveness	[98]
6	Solid lipid nanoparticles	Silybin	Type 2 diabetes	Increased silybin absorption upon oral ingestion	[99]
		Myricetin	Anticancer activity	Significant rise in the proportion of necrosis	

## Data Availability

No new data were created or analyzed in this study. Data sharing is not applicable to this article.

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
