# Peer review of "Exploring the Biomedical Frontiers of Plant-Derived Nanoparticles: Synthesis and Biological Reactions"

_pharmaceutics, 2024, doi:10.3390/pharmaceutics16070923_

Round 1
Reviewer 1 Report
Comments and Suggestions for Authors
The manuscript written by Barathi et al. discusses a topic of interest in recent years, the plant-derived nanoparticles, focusing on their biomedical applications. This topic was intensively studied and many detailed reviews were already published, so finding an original approach is very important. Even though the present manuscript is well documented taking into consideration the number of references cited (118) and the novelty of information (most of the references were published in the last 5 years), I have some major concerns regarding the organization of the information, concerns that should be solved prior to publication.
Major concerns:
1. Abstract: this section should be rewritten according to the changes performed in the revised manuscript and to eliminate the mentions about Cu O, Se, SnO2 nanoparticles that were not found in the manuscript.
2. Introduction: The information presented in this section is too general considering the topic of the review, so I would recommend to replace the introduction part with the 2nd section of the review entitled “Combination of plant extract and nanoparticles” that also presents general aspects. The last paragraph of the introduction, page 2, lines 86-95 can be kept and included in the 2nd section.
3. Cancer therapy: In the lines 140-193, the authors made no mention to plant-derived nanoparticles so it is unclear if the text refers to nanoparticles/metallic nanoparticles in general or to the plant-derived ones, so I would suggest the mention of plant-derived nanoparticles in this section. As regards Table 1, since is entitled “A list of studies that inhibited cancer cell lines using plant extracts in the formation of nanoparticles” I would recommend the introduction within the table of the following columns: the method of obtention of the plant-derived nanoparticles, the cancer cell lines used as in vitro models in the cited studies and a brief description of the potential anticancer mechanism described. A short description of the information presented in the table should be also included in this section
Lines 170-193: no references were cited in these paragraphs, so I recommend to the authors to cite the information presented.
4. In the following sections: Alzheimer's disease, Diabetes and Mosquito-Borne Disease, the authors attention was focused more on describing the diseases than on presenting plant-derived nanoparticles as options of treatment (the main topic of the present review), so I suggest that these sections be improved by adding some more examples of plant-based nanoparticles analyzed as potential treatments for these illnesses.
5. For the improvement of the Drug delivery system section, I recommend the introduction of a table that should include as columns: Type of drug delivery and examples of plant-derived nanoparticles used as such drug delivery system.
6. The section “Advances in plant-based nanotechnology applications are needed in the biomedical field” should be included in the “Future directions” section.
7. The section “Potential process of synthesizing nanoparticles” should be included in “Cancer therapy” section since describes the potential anticancer mechanisms of action of plant-derived nanoparticles.
8. The word “biofate” “encompasses all relevant bio-nano interactions including adsorption of proteins and formation of the protein corona, biodistribution, cellular uptake and subcellular trafficking, drug release, pharmacokinetics, particokinetics, dissociation of the vehicles, and degradation of the constituting materials.” so I would suggest to the authors to either include some details regarding these aspects or removed it from the title of section 6 “Green manufactured nanoparticles' toxicity and biofate”.
9. The Conclusions section should be rewritten since the final phrase refers to chronic respiratory disease and this subject was not discussed in the review.
Minor concerns:
1. Abbreviations should be explained at their first use in the text.
2. The authors used many terms to describe plant-derived nanoparticles, as plant-derived nanoparticles, plant-based nanoparticles, hybrids of plant extracts and nanoparticles, so I suggest that it should be used a single term to avoid confusion
3. In some parts of the manuscript the references are cited as Vancouver, whereas in others as Harvard style. The authors should verify the guidelines of the journal and cite the references within the text accordingly.
Comments on the Quality of English Language
A thorough examination of English grammar and spelling is required since there are some phrases that are difficult to comprehend due to grammar errors:
“These limitations in the accessibility of natural products active components.” Lines 42-43
“ Here, how plant-based nanoparticles are being explored for major disease treatments” -lines 137-138,
and others in the same way.
Author Response
Detailed responses to the Reviewers’ comments
Thank you very much for your kind comments. We will try our best to revise the manuscript in order to improve its quality according to your suggestions and suit the requirements of the journal. Below are the itemized responses from authors to the comments of editor/reviewers (BLUE – Comments; Black – Authors response; Red – Revised text).
Reviewer #1:
The manuscript written by Barathi et al. discusses a topic of interest in recent years, the plant-derived nanoparticles, focusing on their biomedical applications. This topic was intensively studied and many detailed reviews were already published, so finding an original approach is very important. Even though the present manuscript is well documented taking into consideration the number of references cited (118) and the novelty of information (most of the references were published in the last 5 years), I have some major concerns regarding the organization of the information, concerns that should be solved prior to publication.
Major concerns:
- Abstract: this section should be rewritten according to the changes performed in the revised manuscript and to eliminate the mentions about Cu O, Se, SnO2 nanoparticles that were not found in the manuscript.
Thanks for your valuable suggestion and comments. Comments are revised in the main manuscript and removed the heavy metal symbols
- Introduction: The information presented in this section is too general considering the topic of the review, so I would recommend to replace the introduction part with the 2nd section of the review entitled “Combination of plant extract and nanoparticles” that also presents general aspects. The last paragraph of the introduction, page 2, lines 86-95 can be kept and included in the 2nd section.
Thanks for your valuable suggestion, that intro part of 2 page, lines 86-95 were added in the 2nd section
- Cancer therapy: In the lines 140-193, the authors made no mention to plant-derived nanoparticles so it is unclear if the text refers to nanoparticles/metallic nanoparticles in general or to the plant-derived ones, so I would suggest the mention of plant-derived nanoparticles in this section. As regards Table 1, since is entitled “A list of studies that inhibited cancer cell lines using plant extracts in the formation of nanoparticles” I would recommend the introduction within the table of the following columns: the method of obtention of the plant-derived nanoparticles, the cancer cell lines used as in vitro models in the cited studies and a brief description of the potential anticancer mechanism described. A short description of the information presented in the table should be also included in this section. Lines 170-193: no references were cited in these paragraphs, so I recommend to the authors to cite the information presented.
Thanks for your comments, all above mentioned comments are revised and replace. Cites of the information are added. Here, I have mentioned number of cancer treatment by plant derived nano methods only.
- In the following sections: Alzheimer's disease, Diabetes and Mosquito-Borne Disease, the authors attention was focused more on describing the diseases than on presenting plant-derived nanoparticles as options of treatment (the main topic of the present review), so I suggest that these sections be improved by adding some more examples of plant-based nanoparticles analyzed as potential treatments for these illnesses.
Thanks for your valuable suggestion and comments and all above mentioned comments are revised
- For the improvement of the Drug delivery system section, I recommend the introduction of a table that should include as columns: Type of drug delivery and examples of plant-derived nanoparticles used as such drug delivery system.
Thanks for your valuable suggestion and comments
- The section “Advances in plant-based nanotechnology applications are needed in the biomedical field” should be included in the “Future directions” section.
Thanks for your valuable suggestion and comments
- The section “Potential process of synthesizing nanoparticles” should be included in “Cancer therapy” section since describes the potential anticancer mechanisms of action of plant-derived nanoparticles.
Yes, I agree with you. But cancer treatment by plant-derived nanotechnology is one of the important method in biomedical application. Here, I explained about the cabability in that particular disease of cancer.
- The word “biofate” “encompasses all relevant bio-nano interactions including adsorption of proteins and formation of the protein corona, biodistribution, cellular uptake and subcellular trafficking, drug release, pharmacokinetics, particokinetics, dissociation of the vehicles, and degradation of the constituting materials.” so I would suggest to the authors to either include some details regarding these aspects or removed it from the title of section 6 “Green manufactured nanoparticles' toxicity and biofate”.
Thanks for your comments and as per your suggestions I have removed that biofate
- The Conclusions section should be rewritten since the final phrase refers to chronic respiratory disease and this subject was not discussed in the review.
Thanks for your valuable comments and i have modified the conclusion
Minor concerns:
- Abbreviations should be explained at their first use in the text.
Thanks for your valuable comments and your comments are revised
- The authors used many terms to describe plant-derived nanoparticles, as plant-derived nanoparticles, plant-based nanoparticles, hybrids of plant extracts and nanoparticles, so I suggest that it should be used a single term to avoid confusion
Corrected all above-mentioned comments
- In some parts of the manuscript the references are cited as Vancouver, whereas in others as Harvard style. The authors should verify the guidelines of the journal and cite the references within the text accordingly.
References are used with journals templates only
Comments on the Quality of English Language
A thorough examination of English grammar and spelling is required since there are some phrases that are difficult to comprehend due to grammar errors:
“These limitations in the accessibility of natural products active components.” Lines 42-43
“ Here, how plant-based nanoparticles are being explored for major disease treatments” -lines 137-138, and others in the same way.
I am sincerely thanks to the reviewer, who gave this minor and major comments. It’s truly helped me to find out my mistakes and improved the manuscript quality

Reviewer 2 Report
Comments and Suggestions for Authors
In my opinion, the presented review of knowledge about nanomaterials synthesized using plant metabolites for biomedical purposes is very current and worthy of interest. Relatively little is currently known about the biomedical properties of nanoparticles, so studies of this type are an important step in the development of this field. Nevertheless, despite several necessary and interesting knowledge, the possibility of publishing this article in such a prestigious journal as "Pharmaceutics" requires improvement in content and editing.
Substantive suggestions:
1. Even though the title of the article does not suggest delving into the content related to the toxicity of nanostructures and theoretically such a topic does not have to be included in the proposed study, I believe that in the case of review articles on a given topic, information related to the possible negative effects of using nanomaterials for medical purposes should be more emphasized. Currently, little is known about the impact of using nanotechnology, and the mentions on this subject unfortunately clearly suggest that medicine is not yet ready for this type of innovation. I think that in addition to the superlatives of using nanotechnology in medicine, the authors should also critically discuss possible side effects.
2. In line 115, the authors use the term "Hybrids of plant extract and nanoparticle"... in my opinion, this is an abuse in this context and calling the use of plant metabolites for the simple reduction of metal ions to free atoms hybrids is incorrect.
3. I believe that there is no need for a separate subchapter 4. Advances in plant-based nanotechnology applications are needed in the biomedical (line 399) since the entire article actually talks about it...
4. The title of chapter 5. Potential process of synthesizing nanoparticles (line 412) does not correspond to the content presented therein. The title suggests that the authors will present the mechanism of formation of nanostructures using plant extracts, while the chapter describes (very interesting!) possible reaction routes of nanomaterials in contact with the cells of a living organism.
5. The entire Conclusion chapter is basically about nothing. It describes the same problem in several ways, i.e. the need for further research on this topic. I believe that it should either be more concise or should be removed because it does not summarize the previous considerations.
Editing suggestions:
1. Something is incorrect in the authors' e-mail addresses and markings **
2. In the abstract (line 22), the authors do not consistently describe the acronyms of nanoparticles and the word GtNPs is an obvious mistake.
3. Missing subscripts (e.g. line 16, 470, 471), extra space (line 562), lack of indentation of some paragraphs at the beginning
4. The graphics are well put together, but I think that the font in the captions spoils the entire visual effect - I think that their visual quality needs to be improved.
5. Table 1 needs to be edited to make it more readable - there are no spaces between the digit and the unit, the column should be widened to fit the entire word, the notation of significant digits should be standardized
6. The paragraph on lines 288 to 304 has different spacing than the rest of the body maniscript.
Author Response
Detailed responses to the Reviewers’ comments
Thank you very much for your kind comments. We will try our best to revise the manuscript in order to improve its quality according to your suggestions and suit the requirements of the journal. Below are the itemized responses from authors to the comments of editor/reviewers (BLUE – Comments; Black – Authors response; Red – Revised text).
Reviewer #2:
In my opinion, the presented review of knowledge about nanomaterials synthesized using plant metabolites for biomedical purposes is very current and worthy of interest. Relatively little is currently known about the biomedical properties of nanoparticles, so studies of this type are an important step in the development of this field. Nevertheless, despite several necessary and interesting knowledge, the possibility of publishing this article in such a prestigious journal as "Pharmaceutics" requires improvement in content and editing.
Substantive suggestions:
- Even though the title of the article does not suggest delving into the content related to the toxicity of nanostructures and theoretically such a topic does not have to be included in the proposed study, I believe that in the case of review articles on a given topic, information related to the possible negative effects of using nanomaterials formedical purposes should be more emphasized.Currently, little is known about the impact of using nanotechnology, and the mentions on this subject unfortunately clearly suggest that medicine is not yet ready for this type of innovation. I think that in addition to the superlatives of using nanotechnology in medicine, the authors should also critically discuss possible side effects.
Thanks for your valuable comments. All the above-mentioned comments are revised
- In line 115, the authors use the term "Hybrids of plant extract and nanoparticle"... in my opinion, this is an abuse in this context and calling the use of plant metabolites for the simple reduction of metal ions to free atoms hybrids is incorrect.
According to the reviewer suggestions, we have modified that term of Hybrid and revised in the main manuscript
- I believe that there is no need for a separate subchapter 4. Advances in plant-based nanotechnology applications are needed in the biomedical (line 399) since the entire article actually talks about it...
Thanks for your valuable comments. I agreed above-mentioned comments, so I removed that sub title
- The title of chapter 5. Potential process of synthesizing nanoparticles (line 412) does not correspond to the content presented therein.The title suggests that the authors will present the mechanism of formation of nanostructures using plant extracts, while the chapter describes (very interesting!) possible reaction routes of nanomaterials in contact with the cells of a living organism.
Thanks for your valuable comments
- The entire Conclusion chapter is basically about nothing.It describes the same problem in several ways, i.e. the need for further research on this topic.I believe that it should either be more concise or should be removed because it does not summarize the previous considerations.
Thanks for your valuable comments, I have removed that sentence, which reviewer mentioned in the comments section.
Editing suggestions:
- Something is incorrect in the authors' e-mail addresses and markings **
Thanks for your corrections, Comments are revised and marked in main manuscript
- In the abstract (line 22), the authors do not consistently describe the acronyms of nanoparticles and the word GtNPs is an obvious mistake.
Thanks for your corrections, Comments are revised and marked in main manuscript
- Missing subscripts (e.g. line 16, 470, 471), extra space (line 562), lack of indentation of some paragraphs at the beginning
Thanks for your valuable comments and all above-mentioned comments are revised
- The graphics are well put together, but I think that the font in the captions spoils the entire visual effect - I think that their visual quality needs to be improved.
Thanks for your valuable comments and all above-mentioned comments are revised
- Table 1 needs to be edited to make it more readable - there are no spaces between the digit and the unit, the column should be widened to fit the entire word, the notation of significant digits should be standardized
Thanks for your valuable comments and all above-mentioned comments are revised
- The paragraph on lines 288 to 304 has different spacing than the rest of the body maniscript.
Thanks for your valuable comments and above-mentioned comments are revised

Reviewer 3 Report
Comments and Suggestions for Authors
This paper reviews recent developments in plant-based nanoparticles for biomedical applications. The direction of the review is scientifically interesting, but the content is superficial, seems immature and lacks scientific validity (with the exception of chapter 6). The manuscript of the current version cannot be recommended for publication.
The manuscript contains stylistic and consistency errors.
Author Response
Detailed responses to the Reviewers’ comments
Thank you very much for your kind comments. We will try our best to revise the manuscript in order to improve its quality according to your suggestions and suit the requirements of the journal. Below are the itemized responses from authors to the comments of editor/reviewers (BLUE – Comments; Black – Authors response; Red – Revised text).
Reviewer #3:
This paper reviews recent developments in plant-based nanoparticles for biomedical applications. The direction of the review is scientifically interesting, but the content is superficial, seems immature and lacks scientific validity (with the exception of chapter 6). The manuscript of the current version cannot be recommended for publication.
Thanks for your valuable comments. All the above-mentioned comments are revised

Reviewer 4 Report
Comments and Suggestions for Authors
The manuscript aims to highlight the use of metal nanoparticles (NPs) such as Pt, CuO, Se, Ag, Au, SnO2 and ZnO of plant origin as drug delivery systems.
Specifically, the authors review the methods of green synthesis and derivation of nanoparticles from plant-derived compounds such as plants, fruits and vegetables, and their application in the treatment of major diseases such as Alzheimer's, cancer, diabetes, etc. The manuscript is well and clearly written and the conclusions are useful for further development of nanoparticles.
The reference list contains all publications, including the most modern ones, dedicated to the topic. The figures are appropriate and clearly present the derivation and application of these nanoparticles.
Author Response
Detailed responses to the Reviewers’ comments
Thank you very much for your kind comments. We will try our best to revise the manuscript in order to improve its quality according to your suggestions and suit the requirements of the journal. Below are the itemized responses from authors to the comments of editor/reviewers (BLUE – Comments; Black – Authors response; Red – Revised text).
Reviewer #4:
The manuscript aims to highlight the use of metal nanoparticles (NPs) such as Pt, CuO, Se, Ag, Au, SnO2 and ZnO of plant origin as drug delivery systems.
Specifically, the authors review the methods of green synthesis and derivation of nanoparticles from plant-derived compounds such as plants, fruits and vegetables, and their application in the treatment of major diseases such as Alzheimer's, cancer, diabetes, etc. The manuscript is well and clearly written and the conclusions are useful for further development of nanoparticles.
The reference list contains all publications, including the most modern ones, dedicated to the topic. The figures are appropriate and clearly present the derivation and application of these nanoparticles.
Thank you so much for your kind comments on my manuscript, it truly encouraged and gave more responses. Once again thanks

Reviewer 5 Report
Comments and Suggestions for Authors
Although this paper introduces the preparation, properties, and biomedical applications of plant-based nanoparticles, the author only discusses some characteristics of the nanoparticles themselves, which are solely related to their physical and chemical properties. However, there is no direct correlation with how these nanoparticles are obtained (whether through plant extract preparation or conventional chemical methods).
1、The title needs to be revised to avoid causing misunderstanding that the nanoparticles are directly extracted from plants. In fact, the nanoparticles are metal and metal oxide nanoparticles prepared using plant extracts.
2、There is a considerable amount of content that is irrelevant to the main theme of this review, such as the section on liposome nanoparticles of andrographolide (AG) (lines 383-398). There are many other similar instances in the article.
3、Many of the advantages discussed in this paper regarding plant-based nanoparticles are not unique; they are shared by other nanoparticles prepared through other methods. The author needs to uncover the truly unique advantages of plant-based nanoparticles.
4、In Chapter 5, "Potential Process of Synthesizing Nanoparticles," it needs to be clarified whether the therapeutic effects of nanoparticles are solely due to the plant extracts used in plant-based nanoparticles or if they are inherent in the metal nanoparticles themselves.
5、In Chapter 6, when discussing the toxicity and biosafety of nanoparticles, it is important to determine whether the toxicity is a characteristic of the nanoparticles themselves or specific to plant-based nanoparticles.
Author Response
Detailed responses to the Reviewers’ comments
Thank you very much for your kind comments. We will try our best to revise the manuscript in order to improve its quality according to your suggestions and suit the requirements of the journal. Below are the itemized responses from authors to the comments of editor/reviewers (BLUE – Comments; Black – Authors response; Red – Revised text).
Reviewer #5:
Although this paper introduces the preparation, properties, and biomedical applications of plant-based nanoparticles, the author only discusses some characteristics of the nanoparticles themselves, which are solely related to their physical and chemical properties. However, there is no direct correlation with how these nanoparticles are obtained (whether through plant extract preparation or conventional chemical methods).
Thanks for your comments and this manuscript is mainly focused on the plant extract preparation only
1、The title needs to be revised to avoid causing misunderstanding that the nanoparticles are directly extracted from plants. In fact, the nanoparticles are metal and metal oxide nanoparticles prepared using plant extracts.
I agreed with your comments and has changed the title of “Exploring the Biomedical Frontiers of Plant-Derived Nanoparticles: Synthesis and Biological Reactions”
2、There is a considerable amount of content that is irrelevant to the main theme of this review, such as the section on liposome nanoparticles of andrographolide (AG) (lines 383-398). There are many other similar instances in the article.
The lines 383-398 are removed from the reviewer suggestion, also read carefully and revised the sentences line by line
3、Many of the advantages discussed in this paper regarding plant-based nanoparticles are not unique; they are shared by other nanoparticles prepared through other methods. The author needs to uncover the truly unique advantages of plant-based nanoparticles.
Thanks for your suggestions, plant-based nanoparticles having several advantages in different fields and our manuscript is focused on the biomedical application with diseases of cancer, Alzheimer, Diabetes, Mosquito-borne disease. According to the recent reviewers informations, I has covered upto my knowledge and I will learn to find that unique advantages in my next paper.
4、In Chapter 5, "Potential Process of Synthesizing Nanoparticles," it needs to be clarified whether the therapeutic effects of nanoparticles are solely due to the plant extracts used in plant-based nanoparticles or if they are inherent in the metal nanoparticles themselves.
Thanks for your valuable comments. For examples, silver nano-particles are naturally having some medicinal properties and it can heal some of the diseases (Cancer) with its minimum level, but when the NPs combined with the plant parts extracts. The healing percentage is very high. Accordingly, both is having therapeutic effects.
5、In Chapter 6, when discussing the toxicity and biosafety of nanoparticles, it is important to determine whether the toxicity is a characteristic of the nanoparticles themselves or specific to plant-based nanoparticles.
Yes, I agree with your comments. As per my knowledge both is having toxicity effects. The combination of plants derived nano-particles are having less toxic nature, when compared with raw nano-particles. It may differ based on the concentrations, plants selection, diseases and health condition. For your reference, here I pasted one article title “Structural parameters of nanoparticles affecting their toxicity for biomedical applications: a review”.

Round 2
Reviewer 1 Report
Comments and Suggestions for Authors
The authors performed most of the changes suggested in my previous report. Still, I have some recommendations in order to improve the revised version of the manuscript, as follows:
1. As mentioned in the my previous report, I would remove the Introduction section and transform the second section into Introduction (similar information in presented in both sections), but if the authors and the Editor consider that this section should to be kept, is their choice to make.
2. The title of table 1 should be changed to "Types of planted-derived nanoparticles that showed anticancer potential"
3. For the improvement of the Drug delivery system section, I recommend the introduction of a table that should include as columns: Type of drug delivery and examples of plant-derived nanoparticles used as such drug delivery system
4. The title of section 4 "Potential process of synthesizing nanoparticles" should be changed to "Potential anticancer mechanism of action of plant-derived nanoparticles" based on the data presented in this section
5. Conclusion section:
- lines 569-570 should be rephrased
- lines 573-576 should be removed - no data was presented in the review about chronic respiratory disease until this point
Comments on the Quality of English Language
A polish of the English grammar and spelling is still recommended.
Author Response
Detailed responses to the Reviewers’ comments
Thank you very much for your kind comments. We will try our best to revise the manuscript in order to improve its quality according to your suggestions and suit the requirements of the journal. Below are the itemized responses from authors to the comments of editor/reviewers (BLUE – Comments; Black – Authors response; Red – Revised text).
Reviewer #1:
The authors performed most of the changes suggested in my previous report. Still, I have some recommendations in order to improve the revised version of the manuscript, as follows:
- As mentioned in my previous report, I would remove the Introduction section and transform the second section into Introduction (similar information in presented in both sections), but if the authors and the Editor consider that this section should to be kept, is their choice to make.
Thanks for your valuable suggestion and comments. We would like to keep this section separately (Combination of plant extract and nanoparticles)
- The title of table 1 should be changed to "Types of planted-derived nanoparticles that showed anticancer potential"
I agree with your comments and thanks for your valuable suggestion and comments are revised.
- For the improvement of the Drug delivery system section, I recommend the introduction of a table that should include as columns: Type of drug delivery and examples of plant-derived nanoparticles used as such drug delivery system
The comments are revised and added in the manuscript as per suggestions
- The title of section 4 "Potential process of synthesizing nanoparticles" should be changed to "Potential anticancer mechanism of action of plant-derived nanoparticles" based on the data presented in this section
Yes. I changed the title and replaced with Potential anticancer mechanism of action of plant-derived nanoparticles
- Conclusion section:
- lines 569-570 should be rephrased
The lines 569-570 has rephrased
- lines 573-576 should be removed - no data was presented in the review about chronic respiratory disease until this point
The mentioned lines are removed as per your suggestion
Reviewer 2 Report
Comments and Suggestions for Authors
I would like to thank the authors for considering all the suggestions and after proofreading, I believe that the manuscript is more interesting to read and contains more logical content. The only issues left for further correction are editorial issues - in lines 65 and 237 there is a missing hyphen next to the words plant-derived, italics in the name of the bacteria (line 237) and in Table 1 The authors alternately use a space or not between the digit and the unit nm - you need to be consistent.
Author Response
Detailed responses to the Reviewers’ comments
Thank you very much for your kind comments. We will try our best to revise the manuscript in order to improve its quality according to your suggestions and suit the requirements of the journal. Below are the itemized responses from authors to the comments of editor/reviewers (BLUE – Comments; Black – Authors response; Red – Revised text).
Reviewer #2:
I would like to thank the authors for considering all the suggestions and after proofreading, I believe that the manuscript is more interesting to read and contains more logical content. The only issues left for further correction are editorial issues - in lines 65 and 237 there is a missing hyphen next to the words plant-derived, italics in the name of the bacteria (line 237) and in Table 1 The authors alternately use a space or not between the digit and the unit nm - you need to be consistent.
Thanks for your valuable suggestions and comments are revised
Reviewer 3 Report
Comments and Suggestions for Authors
Thanks to the authors for making changes to the manuscript. The overall quality has. Some points:
- Nanoparticle abbreviations should be written in one style (line 22-23).
- Table 1 should also include an abbreviation for silver nanoparticles
- Is it necessary to state "The purpose of the current reviews is to determine how plant extracts contribute to the synthesis of Silver nanoparticles (AgNPs), Gold nanoparticles (GtNPs), and platinum nanoparticles (PtNPs)" in the Abstract when almost all the chapters of the manuscript mention only "plant-derived nanoparticles".
Comments on the Quality of English LanguageThere are some errors and inaccuracies.
Author Response
Detailed responses to the Reviewers’ comments
Thank you very much for your kind comments. We will try our best to revise the manuscript in order to improve its quality according to your suggestions and suit the requirements of the journal. Below are the itemized responses from authors to the comments of editor/reviewers (BLUE – Comments; Black – Authors response; Red – Revised text).
Reviewer #3:
Nanoparticle abbreviations should be written in one style (line 22-23).
Thanks for your comments and the mentioned comments are revised
Table 1 should also include an abbreviation for silver nanoparticles
Thanks for your comments and the mentioned comments are revised
Is it necessary to state "The purpose of the current reviews is to determine how plant extracts contribute to the synthesis of Silver nanoparticles (AgNPs), Gold nanoparticles (GtNPs), and platinum nanoparticles (PtNPs)" in the Abstract when almost all the chapters of the manuscript mention only "plant-derived nanoparticles".
Yes, I agree with your comments and for the examples purpose only we have added here.
Reviewer 5 Report
Comments and Suggestions for Authors
after this revision, this work could be accepted.
Comments on the Quality of English LanguageMinor editing of English language is required
Author Response
Detailed responses to the Reviewers’ comments
Thank you very much for your kind comments. We will try our best to revise the manuscript in order to improve its quality according to your suggestions and suit the requirements of the journal. Below are the itemized responses from authors to the comments of editor/reviewers (BLUE – Comments; Black – Authors response; Red – Revised text).
Thanks for your comments and the mentioned comments are revised